# Green Veterinary Pharmacology Applied to Parasite Control: Evaluation of *Punica granatum*, *Artemisia campestris*, *Salix caprea* Aqueous Macerates against Gastrointestinal Nematodes of Sheep

**DOI:** 10.3390/vetsci8100237

**Published:** 2021-10-15

**Authors:** Fabio Castagna, Cristian Piras, Ernesto Palma, Vincenzo Musolino, Carmine Lupia, Antonio Bosco, Laura Rinaldi, Giuseppe Cringoli, Vincenzo Musella, Domenico Britti

**Affiliations:** 1Department of Health Sciences, University of Catanzaro Magna Græcia, CISVetSUA, 88100 Catanzaro, Italy; c.piras@unicz.it (C.P.); palma@unicz.it (E.P.); studiolupiacarmine@libero.it (C.L.); musella@unicz.it (V.M.); britti@unicz.it (D.B.); 2Nutramed S.c.a.r.l. Complesso Ninì Barbieri, Roccelletta di Borgia, 88021 Catanzaro, Italy; 3Department of Health Sciences, Institute of Research for Food Safety & Health (IRC-FISH), University of Catanzaro Magna Græcia, 88100 Catanzaro, Italy; 4Mediterranean Etnobotanical Conservatory, Sersale (CZ), 88054 Catanzaro, Italy; 5Department of Veterinary Medicine and Animal Production, University of Naples Federico II, CREMOPAR Regione Campania, 80137 Naples, Italy; boscoant@tiscali.it (A.B.); lrinaldi@unina.it (L.R.); cringoli@unina.it (G.C.)

**Keywords:** *Punica granatum*, *Artemisia campestris*, *Salix caprea*, gastrointestinal nematodes, anthelmintic efficacy, sheep, ethnoveterinary medicine, green veterinary pharmacology

## Abstract

Resistance to anthelmintic drugs in gastrointestinal nematodes (GIN) of sheep is of high concern for livestock production worldwide. In Calabria (southern Italy), many plants have been used in ethnoveterinary medicine for parasite control in small ruminants. Here, we present an in vivo evaluation of anthelmintic efficacy of three plant extracts. The first was based on bark and leaves of *Salix caprea*, the second and the third were based on the whole plant *Artemisia campestris* and whole fruit (seeds and peel) of *Punica granatum*, respectively. Anthelmintic efficacy was evaluated according to the fecal egg count reduction test (FECRT) performed with the FLOTAC technique. The results showed a significant anthelmintic effect of *Punica granatum* macerate (50%), a low effectiveness of the *Artemisia campestris* macerate (20%), and a complete ineffectiveness of *Salix caprea* macerate (0.1%). With these outcomes, we report a *P. granatum*-based remedy reducing 50% GIN egg output. This result was obtained without using any synthetic drug, paving the way for the employment of green veterinary pharmacology (GVP) as a complementary and sustainable method to reduce the use of chemicals and to counteract anthelmintic resistance.

## 1. Introduction

Sheep and goat animal productions are particularly important in the European area, in which are reared 16.8 million goats and 130.8 million sheep. This ensures livelihoods for vulnerable populations in rural areas, including those living in under-developed areas [1]. Small ruminant dairying is of major importance for the agricultural economy of the Mediterranean basin [2], where extensive grazing-based ruminant systems have a long tradition dating back to antiquity [3].

In Italy, a lot of sheep and goats are bred in several different conditions due to the huge variety of pasture environments. In the Calabria region, sheep farming represents an important economic resource for the agri-food sector, particularly for local dairy producers [4]. Traditional sheep dairy products include D.O.P. (Protected Designation of Origin) and P.A.T. (Italian traditional agri-food products) cheeses. Besides milk production, sheep and goat farming plays an important role for the preservation and shaping of mountainous landscapes, especially at high altitudes and steep slopes [5]. In addition, small ruminants dairying represents, for this territory, a reality that has a value that goes well beyond its economic purpose if we consider the important role in keeping ancient cultures and traditions in rural areas.

However, this livestock production system is threatened by infectious and parasitic diseases that negatively affect the quantity and the quality of production. Parasitic infections are of economic importance worldwide and cause diminished weight gains, decreased milk yields, and discarded organs at slaughter [6,7].

In particular, gastrointestinal nematode (GIN) infection (caused by different genera of nematodes, e.g., *Teladorsagia*, *Haemonchus*, *Trichostrongylus*, and *Oesophagostomum*) remains one of the main constraints to small ruminant production in southern Italy [8]. The treatments of GIN infection in sheep is almost exclusively based on the use of synthetic drugs [4]. However, the improper use (over-and mis-use) of anthelmintics has led to development of anthelmintic resistance (AR) [9]. AR to one or more drug classes in GIN species is widely distributed, especially in small ruminants [10]. This phenomenon is of high concern for the livestock industry in the most industrialized countries, recently becoming a threat also in Italy [11]. AR is still rare in southern Italy, which is due to some concrete actions, such as the constant monitoring of GIN infection by regular diagnosis, the use of targeted treatments, the alternation of different drugs, the correct drenching, and the minimization of animals movement between farms [12]. However, AR was recently detected in two farms from an area considered anthelmintic resistance-free in southern Italy [9]. Therefore, a constant monitoring of the efficacy of anthelmintics in sheep in southern Italy is strongly recommended [9,12]. Recent studies have estimated the annual cost deriving from AR in Europe in €38 million and this cost could be increased by the growing spread of helminth populations resistant to multiple anthelmintic drug classes [13].

It is therefore necessary to identify alternative (or at least complementary) and sustainable practices for GIN control to minimize the use of chemicals and to counteract AR. These approaches would promote the animal, the environment, and consumer health [14]. In particular, plants’ therapeutic properties may be used in ethnoveterinary practices against GIN and would provide successful alternative remedies to synthetic drugs. From this perspective, ethnoveterinary medicine, which has been transferred from generation to generation, represents an integral part of medical practices in many countries [15,16,17,18,19,20]. Since this practice is very different between geographical regions, this information is difficult to retrieve and requires some preliminary work in order to identify the possible treatments of major importance [21].

According to this premise, we decided to perform a preliminary screening based on interviews and questionnaires to collect the knowledge about the most used ethno-veterinary remedies. The most promising and ease-to-use approaches were subsequently “in vivo” tested in sheep farms using standardized conditions [22]. Among the possible plant macerates to be possibly employed, there were *Punica granatum*, *Artemisia campestris*, and *Salix caprea.*

*Punica**granatum* L. (Lythraceae), commonly known as pomegranate, is a deciduous shrub or tree (2–6 m in height) of genus *Punica*, belonging to Lythraceae family. It has many spiny branches. The leaves are glossy, lance-shaped, or ovate, leathery generally rounded to the apex. Flowers are subsessile; calyx red, leathery. Petals 5–6, red, rarely red orange. Fruit is up to 6–12 cm wide with a deep-red, leathery skin. The fruit, balausta, is often considered to be a large berry. The tough, leathery skin is typically yellow overlaid with pink or red. The fruit has numerous seeds, rich in oil, and they are located in the fruit, separated by a white, membranous pericarp. Each seed is covered by a gelatinous pulp [23]. It is native to Iran and northern India, but it has spread throughout the Mediterranean region, where it grows from the sea level to 800 m. *P. granatum* L. has been used extensively in the folk medicine of many cultures as a remedy for inflammation, intestinal worms, persistent coughs, diarrhea, and dysentery. Recent findings suggested that *P. granatum* L. might have health-promoting effects mainly through its polyphenol content and antioxidant activity. Fruits, seeds, peel, and leaves of *P. granatum* L. are rich in bioactive compounds that have shown a therapeutic role in the disease’s treatment. *P. granatum* L. peel is a rich source of tannins, flavonoids, and other phenolic compounds. Also, its juice contains active compounds like polyphenols, tannins, anthocyanins, including vitamin C, tocopherols, lipoic acid, and punicalagin, bioactive constituents responsible for more than 50% of the antioxidant activity of pomegranate juice [24]. The anti-parasitic properties of this plant have been previously documented both in vitro [25,26] and in vivo [27] in animal models, and in its extract was documented the presence of compounds gallic acid and ellagic acid, which might explain its anthelmintic efficacy. However, its in vivo efficacy in small ruminants, still needs to be tested.

*Artemisia campestris* L. (Asteraceae), commonly known as field wormwood, is a perennial scarcely aromatic herb or subshrub, that may reach 15–150 cm in height. Its stem is erect and woody in the inferior part. The ascending branches are arched, grooved, brownish, and hairless. The leaves are generally convoluted, with a greyish, glabrescent, cartilaginous surface. The lower leaves are 2–3 pinnatisect, petiolate; the upper, that occur in the inflorescence, are most simple. Inflorescences axillary, racemose bracts very small. Capitula are pear-shaped or globose usually shortly pedunculate, erect or erecto-patent, rarely recurved, containing 8 to 12 flowers, organized on convex and glabrous receptacle. The ray flowers, female, pistillate, and fertile; the disk flowers, males, are sterile with reduced abortive ovaries [23]. *A. campestris* L, growing in dry places in most of Europe, and in Italy found [28] at different altitudes up to 2600 m. *A. campestris* L. is abundant in fatty acids, phenolic acids, coumarins, isocoumarins, flavonoids, as well as monoterpenes and sesquiterpenes in essential oils. In ethnomedicine, the leaves, steams, flowers, and aerial part of the plant are decocted, infused, or minced to make a poultice and used as anthelmintic, antidiabetic, antihypertensive, emmenagogue, and antivenom to treat digestive and cutaneous problems [29]. This plant’s essential oil has the compounds Limonene, Beta-Pinene, and many others that might contribute to its anthelmintic efficacy [30]; however, the extraction of the essential oil makes the procedure time and cost inconvenient to be used in large-scale small ruminants treatments.

*Salix caprea* L. (Salicaceae), commonly known as goat willow, is a deciduous medium-size tree or shrub up to 10 m meters in height. The stems are small, with a gray bark irregularly and grossly cracked. The twigs are greenish, thick, and with grey hairs, becoming glabrescent. The leaves are alternate, broadly elliptic, and up to 5–12 cm long. The upper side of the leaf is green, opaque, glabrescent, while the underside is covered by densely and soft white-downy hairs. *S. caprea* L. dioecious species and aments, also named catkins, appear in March or April. Male catkins have spreading yellow stamens, while female catkins are greenish and insect pollinated. The catkins have 100–200 flowers in each female catkin and 200–300 flowers in each male catkin pulp [23]. Although, this species is native to cool temperate and boreal regions of Europe and Asia, it has spread throughout the Mediterranean region. *S. caprea* L. is a rich source of tannins like flavonoids, glycosides, procyanidins, organic acids, and their derivatives, sterols, terpenes, and fatty acids were reported. *Salix* leaves mainly contain flavonoids, phenolic acids, their derivatives, and phenolic glycosides, while stem bark mainly contains procyanidins. In traditional medicine, goat willow extracts are used as antiseptic, eye tonic, painkiller, astringent, or even to treat malaria, gout, neuralgia, and intestinal diseases [31]. *Salix* genus contains both phenolic compounds and tannins that showed an in vitro anthelmintic efficacy [32]. However, even in this case, as for *Artemisia*, the anthelmintic efficacy was tested after the extraction of the bioactive compounds.

According to this premise, we decided to pursue the green veterinary pharmacology (GVP) aim by “in vivo” analyzing the anthelmintic efficacy of these plants in order to identify some alternative remedies useful for the control of AR phenomena. The plant here investigated for the in vivo anthelmintic efficacy are prepared in field using a simple water maceration/extraction procedure.

## 2. Materials and Methods

### 2.1. Surveys and Ethnoveterinary Remedies Identification

All farmers (105, distributed in the province of Catanzaro, Calabria, Italy) were asked to fill a questionnaire about their practices of farm management, such as: the husbandry system, breeds, and number of animals, size of pastures, and worm-control practices (parasitological exams, treatment times and frequency, products, and dosages of the anthelmintic drugs). In addition, preventive measures against parasites were examined, such as the choice of anthelmintics, application practices, the perceived effectiveness and side effects. Subsequently, according to the analysis of ethnobotanical data present in the scientific literature and to the newly acquired knowledge, some remedies were enrolled for a comparative analysis to assess their in vivo efficacy against GINs in standardized conditions.

### 2.2. Plant Extract Preparation and Analysis

*Punica granatum* (voucher 114, Mediterranean Etnobotanical Conservatory, Sersale (CZ), Italy);

A total of 20 kg of entire fruits (divided in 4 parts) were macerated in 60 of liters previously boiled water. The maceration period lasted for 10 months. All the fruits were collected during the maximum ripening phase in October. All the fruits grew at 800 m above sea level. The macerate was filtered with a cotton filter and subsequently stored at 16° until use.

*Artemisia campestris* (voucher 115, Mediterranean Etnobotanical Conservatory, Sersale (CZ), Italy) (whole plant and flowers) obtained by maceration in water;

A total of 50 kg of the plant parts, excluding the roots, with around 30% of its blooms (at the beginning of flowering period) were macerated with 70 L of previously boiled water for 30 days. During maceration period, the plants were kept underneath the water level, avoiding air contact. The macerate was filtered and stored at 16° until use.

*Salix caprea* (voucher 116, Mediterranean Etnobotanical Conservatory, Sersale (CZ), Italy) (bark and leaves);

A total of 50 kg of the plant parts, excluding the roots, were macerated with 70 L of water (previously boiled) for 30 days. The plant parts, including bark and leaves were collected at around 800 m above sea level during august. After the end of the maceration process, the mixture was completely extracted by pressing the solid parts and the entire extract was filtered and stored at 16° until use.

All the macerates were prepared by a Calabrian elderly farmer, with plants present in the territory, according to ancient ethnoveterinary recipes handed down for centuries from generation to generation. The macerates were not concentrated, but just filtered and the average yield was of around 70% of the entire starting amount. The only effective macerate (*Punica granatum*) was analyzed using liquid chromatography electrospray ionization mass spectrometry (LC/MS-ESI) as previously described [25].

All the species were collected around Catanzaro (Calabria, Italy) and the taxonomic identification was confirmed by Dr. V. Musolino and Dr. C. Lupia, Department of Health Sciences, University “Magna Graecia” of Catanzaro. Voucher specimens were deposited in the Mediterranean Ethnobotanical Conservatory (Sersale, Catanzaro, Italy) under the following accession numbers: *P. granatum*:114; *A. campestris*: 115; *S. caprea*: 116.

### 2.3. Animals, Faecal Sampling, and Ethnoveterinary Remedies Administration

The animals used for this study were mainly the Sarda breed, homogeneous for age (2 years ± 0.5), body weight (42 Kg ± 1.8) and grazing season, without any anthelmintic treatments for at least 6 months.

Seven days before the study (D-7), individual fecal samples were collected from 120 sheep naturally infected by GIN.

From this parasitological screening, it was possible to select the sheep naturally infected by GINs homogeneous for parasitic intensity expressed in eggs per gram of feces (EPG) for the study.

In particular, a total of 60 sheep were selected and divided into 4 groups of 15 animals each:Treated group 1 (TG1) treated *per OS* (P.O. i.e., oral administration) with 50 mL of *P. granatum* macerate as single dose;Treated group 2 (TG2) treated with 50 mL/P.O. of *A. campestris* macerate as single dose;Treated group 3 (TG3) treated with 50 mL/P.O. of *S. caprea* macerate as single dose;Control group (CG) untreated.

In these studies, the time was: Day 0 (D0) allocation to groups, fecal sampling, and treatment were performed; Days 7, 14, 21 (D7, D14, D21) fecal samples were collected and examined to evaluate anthelmintic efficacy.

All the individual fecal samples (about 10 g) were collected directly from the rectal ampulla of the naturally infected sheep (control and treated experimental groups) in farm.

### 2.4. Parasitological Studies and Evaluation Anthelmintic Efficacy

The individual GIN fecal egg count (FEC) was determined with the FLOTAC technique (made by the University of Naples Federico II), using a sodium chloride flotation solution (specific gravity = 1.200) and a detection limit of 6 eggs per gram of faeces (EPG) [33].

In addition, to identify the GIN genera, the same quantity of feces was collected from each sample to create a pool for each fecal culture group at D0, D7, D14, and D21, following the protocol described by the Ministry of Agriculture, Fisheries and Food [34]. Developed third-stage larvae (L3) were identified using the morphological keys proposed by van Wyk and Mayhew [35]. Identification and percentages of each nematode genera were conducted on 100 L3; if a sample had 100 or less L3 present, all larvae were identified. Thus, on the total number of larvae identified, it was possible to give the percentage of each genus.

Fecal egg count reductions test (FECRT) was used to determine the anthelmintic efficacy. On each fecal sampling day, arithmetic mean EPG was calculated as recommended by the World Association for the Advancement of Veterinary Parasitology (WAAVP) guidelines for evaluating the efficacy of anthelmintics in ruminants, and for each treatment group, percent efficacy (%) was calculated in terms of fecal egg count reduction FECR on the different days [22].

The formula used to evaluate the anthelmintic efficacy (based on the arithmetic mean of the control and treated group) was FECR = 100 × (1 − [T2/C2]), where T2 represents the post-treatment FEC of the treated group, and C2 represents the mean post-treatment FEC of an untreated control group [22]. The statistical analysis for the t-student test was performed with excel and the box plots were obtained with jmpSAS version 16 (SAS Institute srl, Via Darwin 20/22, 20143 Milano ITALIA).

## 3. Results

### 3.1. Surveys and Ethnoveterinary Remedy Identification and Characterization

In the preliminary screening phase, 105 farms were enrolled in the study. All farms identified were specialized in milk production and practiced a semi-extensive breeding system with turn out between March/April–October/November. In the colder months, the animals were housed in stables.

A total of 82% of the farms used rotational pastures and 18% kept the flocks on the same pastures.

It is estimated that over 83% of farmers have used anthelmintics, but only 22% made parasitological diagnosis before treatment. In addition, 20% of farmers were used to treating their animals only upon the appearance of clinical symptoms such as diarrhea, apathy, nasal discharge, and weight loss.

The most commonly used anthelmintic classes were macrocyclic lactones (mainly ivermectin) and benzimidazoles (mainly albendazole and netobimin). Only in seven sheep farms (7.95%), were natural mixtures used.

Of these, seven farms were managed by elderly farmers working and living in these areas according to ancient traditions. These seven farms were using anthelmintic therapies based on natural remedies.

In particular, in two farms, a complementary natural feed based on extracts and essential oils of herbs belonging to the Compositae, Cesalpinacae, Liliacae, Bromeliaceae, and Labiatae families was used to control nematodes and trematodes. This product, which is commercially available and registered for the treatment of different genera of sheep nematodes, trematodes, and coccidia, has been shown to be ineffective in in vivo studies [4,36]. One farmer used an aqueous macerate based on *Dryopteris filix-mas* to control tapeworms; in another farm they integrated dried lupine seeds *(Lupinus albus)* in sheep diet. Finally, another farmer used aqueous macerates of plants against GIN infection. These macerates were produced with plants, or parts of them, present in the study area. The plants were *Artemisia campestris*, *Salix caprea*, and *Punica granatum.* According to the interviews carried out in the preliminary phase, these three candidates emerged as the most valuable to be enrolled in further testing.

### 3.2. Parasitological Studies and Anthelmintic Efficacy

*Punica granatum*, *Artemisia campestris* and *Salix caprea* aqueous macerates were tested for their efficacy in limiting the GINs infestation in sheep.

The percentages of each genera of nematodes present in all sheep groups are reported in Table 1. The percentages of the different GIN genera remained constant before and after treatments.

Table 2 shows the results concerning anthelmintic effectiveness of the aforementioned aqueous vegetable macerates after a single administration. The table shows the GIN EPG (mean) of the different groups and the FECR (%) at different days (D) after treatment.

The GIN EPG values were similar for each experimental group at D0 (Figure 1a). Seven days after the treatment (Figure 1b), a FECR of 50% was observed in the *Punica granatum* group in comparison with the control group (*p* ≤ 0.001). The effect persisted at D14 (Figure 1c) and D21 with a FECR of 44.3% (*p* ≤ 0.05) and 40.4% (*p* ≤ 0.001), respectively. The effect of the *Punica granatum* extract is detailed in Figure 2b where it is possible to see a consistent decrease of EPG in all timepoints in comparison to the control group (Figure 2a). The reduction in EPG was detected after seven days from the treatment (*p* ≤ 0.001) and still, after 14 days, remained lower in comparison to D0 (*p* ≤ 0.05). After 21 days following the treatment, the FEC was similar to D0 (*p* = 0.83); however, as previously described, the EPG reduction was relevant if compared with the EPG of the control group at the same time (Figure 1d, *p* ≤ 0.001). On the contrary, a very low FECR% was observed in the other groups, namely TG2 (*A. campestris)* and TG3 (*S. caprea)*. The most effective *P. granatum* macerate, which was previously in vitro tested against GINs egg hatch test (EHT) [25], was as well chemically characterized for its composition as resumed in Table 3 presented in Section 4.

## 4. Discussion

Livestock sector in Europe is under scrutiny and is challenged by sustainability issues [1], also due to the increasing phenomena of AR and the risk of anthelmintic residues in the environment. In particular, the small ruminant sector should adopt more sustainable practices and principles in order to become more resilient and competitive. The improvement of sustainability represents the center of the debate on the future of global agriculture [1]. Therefore, the increasing diffusion of AR in nematode populations, the risk of drug residues in the environment, the availability and high costs of anthelmintic treatments, especially for low-income farmers, have shown that sustainable helminth control cannot be achieved exclusively with the use of the commercial anthelmintic [47].

In this context, it is necessary to explore and validate alternative/complementary solutions for a sustainable GIN control in small ruminants. Natural compounds and plant extracts are a promising alternative in this direction. Evidence of the anthelmintic properties of plants and plant extracts is derived primarily from ethnoveterinary sources. The use of ethnoveterinary plant preparations has been documented in different parts of the world [17] and these products have, in several cases, shown significant anthelmintic effects against GIN of sheep [48]. In Europe, a relevant number of plants are used to treat organic livestock [20]. Also, in Italy, many plants were used to treat cattle, sheep, poultry, horses, and pigs [49] and some of these plants may be effective against helminth infections in animals [50].

These traditions have survived in some Italian regions [51,52,53] and from the ethno-botanical point of view, the Calabria region of southern Italy represents one of the most valuable areas [54]. In this region, many plants have been used as nutraceuticals [55,56,57,58], in folk medicine [59,60,61], and ethnoveterinary practices [54,61,62,63,64]. However, despite the use of these mixtures, only a few were validated for their use in the veterinary field.

The study was carried out in the Calabria region, in semi-extensive small ruminant farms raised on the pastures of the Ionian side of the Province of Catanzaro (mean altitude 398 mt a.s.l.), with a typically Mediterranean climate.

Calabria region occupies the southern part of the Italian peninsula, it has an area of 15,080 km^2^ and a coastline of 738 km on the Ionian and Tyrrhenian seas. It is one of the most mountainous regions in Italy: 42% of the land is mountainous (elevation < 500 m above sea level (a.s.l.)), 49% hilly (elevation between 50 and 500 m a.s.l.), and only 9% is flat (elevation < 50 m a.s.l.) [65]. Because of its geographic position and mountainous nature, Calabria has a high climatic variability [66] with a typically dry summer subtropical climate, also known as the Mediterranean climate. Coastal zones are characterized by mild winters and hot summers with little precipitation. In particular, the Ionian side, which is influenced by currents coming from Africa, has high temperatures with short and heavy precipitation [67].

The province of Catanzaro is one of the five provinces of the Calabria region. This province has a total area of 5200 square kilometers (2000 sq mi). It occupies the central part of Calabria and is bordered to the west by the Tyrrhenian Sea, to the north by the Sila, to the east by the Ionian Sea, to the south by the Calabrian Serre. In this area, small ruminant farming is very widespread, mostly in marginal areas unsuitable for agricultural production [4]. It is exactly in these areas that traditional ethnoveterinary medicine still survives and is still used for GINs control.

Our study is located in the hot spot of this topic and demonstrates that *A. campestris* and *S. caprea* aqueous macerates used in vivo in sheep slightly reduced GIN EPG, with FECR% between 20.4 and 4.3, and 0.2 and 0.3, at D7 and D21. In contrast, the aqueous pomegranate macerate administered in a single dose showed high FECR% as high as 50.1 at D7 and 40.4 at D21. This effect could be attributed to the synergistic action of the tannin-derivatives and phenolic acids. The chemical characterization with the resumed literature about the potential effects of the detected compounds is resumed in the following Table 3. For each compound, in the last column, are indicated the references with the most relevant papers where the compound might be involved in preparations with anthelmintic functions.

The detected components were mainly alkaloids, tannins, flavonoids, glycosides, and phenols, and many of those, like gallic and ellagic acid, were identified in previous studies of pomegranate parts [68]. This suggests that the main bioactive compounds arise from the fruit properties and not from the maceration process.

Several studies reported that these compounds, present in the different parts of *P. granatum* fruit and plant, have anti-parasitic properties, such as, antiprotozoal activity [69,70,71], anticestodal [72], antinematodal [25,26], and antitrematodal [73,74] effects. Most of the research related to the therapeutic properties of pomegranate is represented by in vitro studies and by a few in vivo studies, regardless of their use in the ethnoveterinary field. The main advantages of using in vitro assays to test the anthelmintic plants are the low costs and rapid turnover, which allow large scale screening of different plants extracts [75]. However, it is important to underline that the concentrations of potentially active substances contained in plants used in vitro do not always correspond to bioavailability in vivo [76]. Although costs of large scale screening of plant extracts/plant products is higher for in vivo studies (FECRT), the latter are more relevant and reliable than in vitro studies [35].

Moreover, the results obtained (Table 1) from the cultures of all groups did not show any significant difference in the ratio between the percentage of genera found pre and post treatment. This result demonstrates that none of the employed treatments are specific for only a genre.

This study confirms what emerged from recent in vitro studies [25], namely that in the Calabria region of southern Italy, there are still some small farmers and shepherds who continue to use traditional vegetable macerates against GIN infections of sheep and that some of them have a significant anthelmintic efficacy, also confirmed by the in vivo tests. It also emerged that the use of these mixtures, the one based on the whole fruit of *P. granatum*, is not reported in the Calabrian ethnoveterinary literature, despite its anti-helminthic efficacy.

The results of these field tests, not only highlight the importance of anthelmintic efficacy studies of ethnoveterinary remedies, but also paves the way for the use of green veterinary pharmacology (GVP) as an alternative and sustainable method to reduce the use of chemicals and to counteract the phenomena of AR.

This proposed solution represents a very important opportunity for farmers, who would have the concrete possibility to change the parasitological control plans in the farms, with many advantages for animal welfare, for the environment, and for overall public health. All this will only be possible by encouraging studies on plants used for GIN control, to identify and validate them, because the major part of these remedies is still encrypted in local traditions persisting from generation to generation. Without further in vivo studies, the ethnopharmacology will remain just “stories and tales” that will be lost over the time.

## 5. Conclusions

The results herein presented, consistently document the effectiveness of natural products, i.e., *Punica granatum* plant macerate of reducing the GIN infections by 50%. Considering the natural origin of this extract and the absence of any documented adverse reactions by treated animals, it is possible to hypothesize the future use of such an ethnoveterinary remedy for sheep nematodes control. Forthcoming anthelmintic resistance phenomena could be approached differently, for example, by employing the described extract as a complementary strategy in parallel to anthelmintic drugs treatment to achieve a synergic effect. Moreover, the production of this extract is easily possible to be done directly in the farm facilities and does not require complex extraction procedures or trained personnel. GVP might represent the easiest future approach to combat AR in livestock ruminants.

## Figures and Tables

**Figure 1 vetsci-08-00237-f001:**
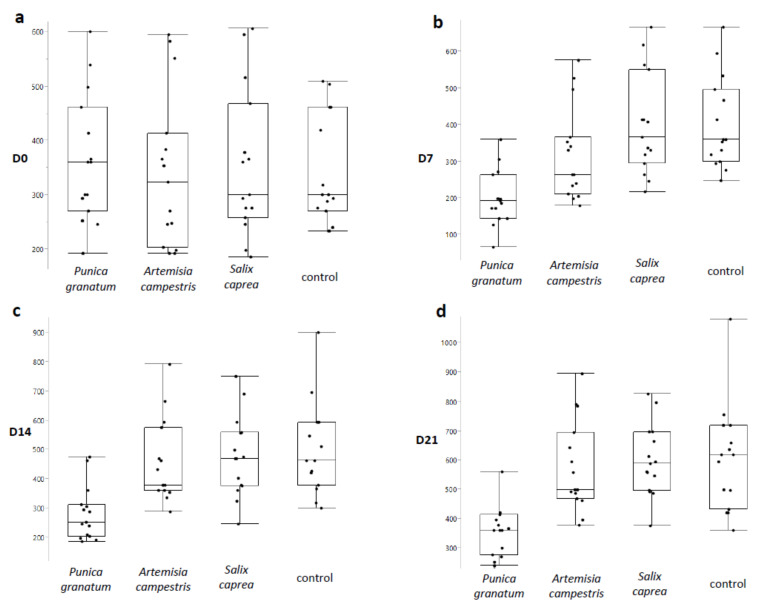
Box plots of gastrointestinal nematodes (GIN) eggs per gram (EPG) at different treatment times [D0 (**a**), D 7 (**b**), D14 (**c**), and D21 (**d**)] in the groups treated with *Punica granatum*, *Artemisia campestris*, *Salix caprea* compared to the untreated control groups.

**Figure 2 vetsci-08-00237-f002:**
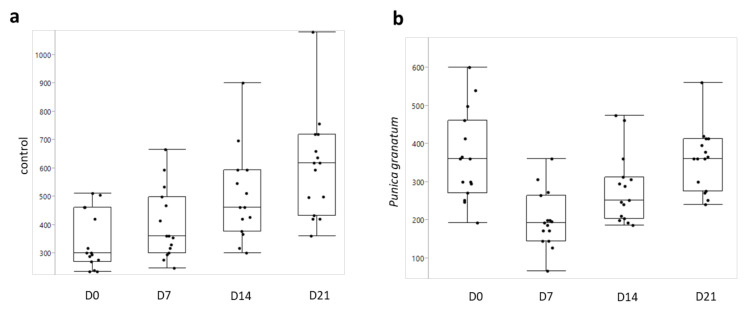
Box plots of GIN EPGs in the control group (**a**) compared to the *Punica granatum* group (**b**) at different times (D0, D7, D14, and D21).

**Table 1 vetsci-08-00237-t001:** Percentage of gastrointestinal nematodes (GIN) third-stage larvae (L3) in each treatment group at D0, D7, D14, and D21. TG1 (Treated group 1); TG2 (Treated group 2); TG3 (Treated group 3); CG (Control group).

Group	Day	*Haemonchus* (%)	*Trichostrongylus* (%)	*Teladorsagia* (%)	*Chabertia* (%)
TG1 *P. granatum*	D0	43	25	26	6
	D7	37	20	34	9
	D14	41	22	31	6
	D21	40	24	28	8
TG2 *A. campestris*	D0	40	29	27	4
	D7	42	27	30	1
	D14	38	29	21	12
	D21	36	28	33	3
TG3 *S. caprea*	D0	45	23	31	1
	D7	37	28	32	3
	D14	42	26	28	4
	D21	40	29	24	7
GG untreated	D0	41	21	31	7
	D7	38	29	32	1
	D14	41	30	28	1
	D21	42	25	28	5

**Table 2 vetsci-08-00237-t002:** Gastrointestinal nematodes (GIN) eggs per gram (EPG) (mean) of the different groups and the fecal egg count reduction (FECR) (%) at different days (D) after treatment; SD (standard deviation).

Groups	D_0_	D_7_		D_14_		D_21_	
	EPG Mean (SD)	EPG Mean (SD)	FECR %	EPG Mean (SD)	FECR %	EPG Mean (SD)	FECR %
TG1 (*P. granatum* group)	363.6 (±117)	199.7 (±74)	50.2	281.6 (±91)	44.3	357.7 (±82)	40.4
TG2 (*A. campestris* group)	341.3 (±141)	319.2 (±126)	20.4	454.8 (±141)	9.8	576.0 (±154)	4.3
TG3 (*S. caprea* group)	354.8 (±134)	400.4 (±139)	0.1	504.0 (±137)	5	599.7 (±121)	0.3
GG (Untreated group)	340.8 (±100)	400.9 (±124)	-	504.4 (±157)	-	601.8 (±181)	-

**Table 3 vetsci-08-00237-t003:** Chemical characterization results of total dry extract. n.d. (not documented).

*m*/*z* Theoretical	*m*/*z* Measured	Analyte	Previously Described Anthelmintic Properties
149.0092	149.0081	Tartaric acid (C_4_H_5_O_6_)	Castagna et al. [25]; Kalaiselvan et al. [37]
181.0718	181.0711	Mannitol (C_6_H_13_0_6_)	Castagna et al. [25]; Cruz-Arévalo et al. [38]
193.0354	193.0347	Glucuronic acid (C_9_ H_9_ O7)	Castagna et al. [25]; Kumar et al. [39]
481.0697	481.0626	2,3-(S)-hexahydroxyphenyl-D-glucose (C_20_H_17_O_14_)	Castagna et al. [25]
169.0142	169.0134	Gallic acid (C_7_H_5_O_5_)	Castagna et al. [25]; Ndjonka et al. [40]; Escareño-Díaz et al. [41]
288.9990	288.9992	Phelligridin J (C_13_H_5_O_8_)	Castagna et al. [25];
469.0049	469.0050	Valoneic acid dilattone (C_21_H_9_O_13_)	Castagna et al. [25]; Khan et al. [42]
197.0455	197.0449	Syringic acid (C_9_H_9_O_5_)	Castagna et al. [25]; Garcia-Bustos et al. [43]; Dkhil et al. [44]
300.9990	300.9991	Ellagic acid (C_14_H_5_O_8_)	Castagna et al. [25]; Mondal et al. [45]; Figueiredo et al. [46]
447.0642	447.0573	Ducheside A (C_20_H_15_O_12_)	Castagna et al. [25]

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
