# Peer review of "Green Veterinary Pharmacology Applied to Parasite Control: Evaluation of Punica granatum, Artemisia campestris, Salix caprea Aqueous Macerates against Gastrointestinal Nematodes of Sheep"

_vetsci, 2021, doi:10.3390/vetsci8100237_

Round 1

Reviewer 1 Report

The article “Green Veterinary Pharmacology applied to parasite control: evaluation of Punica granatum, Artemisia campestris, Salix caprea aqueous macerates in ethnoveterinary medicine against gastrointestinal nematodes of sheep” is an excellent initiative to spread knowledge ethnoveterinary medicine of Calabria region. The work needs to reorganize the introduction and significantly improve the methodology. The goal of the manuscript is unclear and is confusing. The chemical analysis of the extracts needs to be added. Due to the need to make structural and methodological changes, I recommend that this work be recused.
Commentaries: 
Title: The title of your manuscript should be concise, specific, and relevant. I suggest you delete the first part and maintain “evaluation of Punica granatum, Artemisia campestris, Salix caprea aqueous macerates in ethnoveterinary medicine against gastrointestinal nematodes of sheep.” This work contains repeated ideas such as “green veterinary pharmacology” and “ethnoveterinary medicine.”

Introduction: 
The introduction should contextualize the problems, including a specific hypothesis, and finally, briefly mention the objective of the work. The work does not present the sequence suggested by the journal standards. The text presents repeated ideas (lines 83-86), short graphs (lines 75-77, 83-86), and the absence of a clear goal (lines 91-93).
Materials and methods: 
The work is not described with sufficient detail to allow others to replicate it. The development stages of the experiment are not well defined. The application of the questionnaire, production of aqueous macerated, preparation of animals, and application of treatments should be separate and well-detailed topics. It is impossible to reproduce the aqueous extract due to lack of information, such as leaf mass, leaf processing, characterization of extracted molecules, the toxicity of the administered macerated [lines 135-137]. 
Chemical analysis needs to be performed and added.
Results:
The results should be concise, accurate descriptions and follow the order proposed in the methodology.

Author Response

Reviewer 1

The article “Green Veterinary Pharmacology applied to parasite control: evaluation of Punica granatum, Artemisia campestris, Salix caprea aqueous macerates in ethnoveterinary medicine against gastrointestinal nematodes of sheep” is an excellent initiative to spread knowledge ethnoveterinary medicine of Calabria region. The work needs to reorganize the introduction and significantly improve the methodology. The goal of the manuscript is unclear and is confusing. The chemical analysis of the extracts needs to be added. Due to the need to make structural and methodological changes, I recommend that this work be recused.

Commentaries:

Response

Many thanks to the referee for this comment that helped with improving the manuscript. The following points have been revised accordingly:

Title: The title of your manuscript should be concise, specific, and relevant. I suggest you delete the first part and maintain “evaluation of Punica granatum, Artemisia campestris, Salix caprea aqueous macerates in ethnoveterinary medicine against gastrointestinal nematodes of sheep.” This work contains repeated ideas such as “green veterinary pharmacology” and “ethnoveterinary medicine.”

Response

All the authors are grateful to the referee for this helpful comment. The title has now been modified to make it more focused. All authors decided to keep the term “Green Veterinary Pharmacology” that embraces a new research thread that is seeking for natural remedies and non-environment-persisting molecules as alternatives for more sustainable pharmacological approaches. However, we are grateful for pointing out that removing “ethnoveterinary medicine” concept makes the title simpler and more concise.

Introduction:

The introduction should contextualize the problems, including a specific hypothesis, and finally, briefly mention the objective of the work. The work does not present the sequence suggested by the journal standards. The text presents repeated ideas (lines 83-86), short graphs (lines 75-77, 83-86), and the absence of a clear goal (lines 91-93).

Response

Many thanks to the referee for this comment. The introduction section has been amended as requested to make it more focused, the objective of the work has now been clearly stated in the end of the introduction section. The template adopted now precisely follows the order suggested by the journal, including the subsections numbering. All repeated ideas have now been removed accordingly. We hope that the work done is now satisfactory for the reviewer.

Materials and methods:

The work is not described with sufficient detail to allow others to replicate it. The development stages of the experiment are not well defined. The application of the questionnaire, production of aqueous macerated, preparation of animals, and application of treatments should be separate and well-detailed topics. It is impossible to reproduce the aqueous extract due to lack of information, such as leaf mass, leaf processing, characterization of extracted molecules, the toxicity of the administered macerated [lines 135-137].

Chemical analysis needs to be performed and added.

Response

Many thanks to the referee for this comment. The methods section has now been implemented with the requested details. The results of the chemical characterization that was performed together with the previous “in vivo” study of the most effective plant (Punica granatum) has now been described in the discussion section.

Results:

The results should be concise, accurate descriptions and follow the order proposed in the methodology.

Response:

Many thanks to the referee for rising this very relevant point. The results section has been amended accordingly. More precisely, we amended the methods section by modifying the methods order that now is consistent with the results section.

Reviewer 2 Report

The manuscript "Green Veterinary Pharmacology applied to parasite control:  evaluation of Punica granatum, Artemisia campestris, Salix caprea aqueous macerates in ethno-veterinary medicine against  gastrointestinal nematodes of sheep" was well done. However, some details should be improved. For instance, in the introduction section lack information about the medicinal properties of the plants studied. It is very important to justify why used this plants in their study.

Other important point, when it perform studies with medicinal plants, its should have an chemical analysis in the order to identify the major secondary compounds of the extracts. Since, that chemical composition of between plan and plant of the same genus is different and other environmental factors are involved on their phytochemical profile. So, why did not carried out an chemical analysis of the extracts?.

This is important; since to standardize the extracts before use to treat the animals is necessary know the composition of the secondary compounds. How do you justify it?

Some comments and suggestions are into the manuscript.

Author Response

Reviewer 2

The manuscript "Green Veterinary Pharmacology applied to parasite control:  evaluation of Punica granatum, Artemisia campestris, Salix caprea aqueous macerates in ethno-veterinary medicine against gastrointestinal nematodes of sheep" was well done. However, some details should be improved. For instance, in the introduction section lack information about the medicinal properties of the plants studied. It is very important to justify why used this plants in their study.

Response

Many thanks to the referee for this comment. The introduction section has been improved, as requested, with a brief description of the plant extracts studied for their anthelmintic activity.

Other important point, when it perform studies with medicinal plants, its should have an chemical analysis in the order to identify the major secondary compounds of the extracts. Since, that chemical composition of between plan and plant of the same genus is different and other environmental factors are involved on their phytochemical profile. So, why did not carried out an chemical analysis of the extracts?.

Response

All authors are particularly grateful to the referee for this comment. At the beginning of the project we decided to adopt the policy to chemically characterize the extracts that were found to be effective “in vivo”. For this reason we were particularly happy to improve the manuscript with the information about the chemical characterization.

This is important; since to standardize the extracts before use to treat the animals is necessary know the composition of the secondary compounds. How do you justify it?

Response

All the manuscript has been revised according to this comment. The chemical characterization of the extract used has now been indicated in the manuscript. This includes both primary and secondary compounds.

Some comments and suggestions are into the manuscript.

Which was the criteria to choose this dose? Please add a reference to support it.

Response

Many thanks to the referee for this comment. Unfortunately, there is no reference available. The amount used was chosen according to the ethnoveterinary field experience of the breeder.

Please, add standard deviation to each mean.

Response

Many thanks to the referee for this comment. SD values now added.

Part of this information is same to the table 2. Consider complement the table 2 with this information (standard deviation). and replace this figure.

Response

Many thanks to the referee for this comment. The table has been improved as requested, however, the figure does not provide redundant information. All the authors agree to leave the figure because it indicates every measurement performed and provides a visual representation of the dataset.

On behalf of all authors, we would like to express our gratitude for the effort and the constructive comments that helped to improve the manuscript. All the minor comments present in the pdf file have been addressed.

Reviewer 3 Report

In this manuscript, the authors address the serious problem of anthelmintic resistance in the veterinary field and advocate the combined use of natural products and anthelmintics to reduce anthelmintic resistance. They test a number of allegedly anthelmintic plant recipes used by sheep farmers in the Calabria region and conclude that the use of a pomegranate fruit macerate (Punica granatum) has relevant anthelmintic effects at least in sheep.

The article is well written, interesting and comprehensive. In the attached pdf version, several comments, generally minor, have been marked for correction of the manuscript.

The References section which needs to be checked thoroughly.

For this reviewer only one relevant issue is raised. The authors do not describe the preparation of the plant extracts used, i.e. the proportion of the different plant parts used, the weight of each part in relation to the volume of water used for the maceration, the conditions of the maceration, etc., which means that the results of this work cannot be used by other farmers in other areas where Punica granatum is present. The authors should clarify these issues which the reviewer believes would add great value to their work.

Author Response

Reviewer 3

In this manuscript, the authors address the serious problem of anthelmintic resistance in the veterinary field and advocate the combined use of natural products and anthelmintics to reduce anthelmintic resistance. They test a number of allegedly anthelmintic plant recipes used by sheep farmers in the Calabria region and conclude that the use of a pomegranate fruit macerate (Punica granatum) has relevant anthelmintic effects at least in sheep.

The article is well written, interesting and comprehensive. In the attached pdf version, several comments, generally minor, have been marked for correction of the manuscript.

The References section which needs to be checked thoroughly.

For this reviewer only one relevant issue is raised. The authors do not describe the preparation of the plant extracts used, i.e. the proportion of the different plant parts used, the weight of each part in relation to the volume of water used for the maceration, the conditions of the maceration, etc., which means that the results of this work cannot be used by other farmers in other areas where Punica granatum is present. The authors should clarify these issues which the reviewer believes would add great value to their work.

Response

Thanks for rising this very important point. We now improved the methods section by trying to be as exhaustive as possible about the preparation of the macerates.

On behalf of all authors, I would like to communicate our gratitude for the effort and the constructive comments that helped to improve the manuscript.

All the minor comments present in the pdf file have been addressed.

Round 2

Reviewer 1 Report

The manuscript increased the quality of the new version. Some points need to be more clear. Below are the specific commentaries: 

The introduction improved; however, information about how the compound detected in these plants could be ATH effect was missed.

Line 141 – GVP was not descript before in the text-only in the abstract.

M&M - How was, did the literature review?

Line 146 – All farmers? From where? How many?

I presume that the description from 167 to 171 is from Punica.

Did botanists identify the plants? Would you please add the voucher number?

Were not grow fungi or bacteria? 30 days in water? How do you affirm that it is from the plant extract and not fungi or bacteria compound? How was the yield of the extract? Have had concentrate on the extract?

Most references from Table 3 didn’t show anthelmintic properties from the compound. Extracts with this compound have been demonstrated with ATH activity. This is entirely different.

Author Response

Comments and Suggestions for Authors

The manuscript increased the quality of the new version. Some points need to be more clear. Below are the specific commentaries:

The introduction improved; however, information about how the compound detected in these plants could be ATH effect was missed.

Response

Thanks for this comment that made the introduction more focused. The missing link between the plants employed in this study and their possible anthelmintic efficacy has now been explained in the introduction section in each pant’s paragraph.

Line 141 – GVP was not descript before in the text-only in the abstract.

Response

Thanks for this comment. Now amended.

M&M - How was, did the literature review?

Response

Thanks for this comment. A new literature review and a bibliography update was performed and reported in the text.

Line 146 – All farmers? From where? How many?

Response

Thanks for this comment. Now specified in the beginning of the material and methods section.

I presume that the description from 167 to 171 is from Punica.

Response

Many thanks for this really spot on comment. The section has now been amended.

Did botanists identify the plants? Would you please add the voucher number?

Response

Yes, the botanists did identify the plants, the corresponding voucher numbers are now indicated in the text in the material and methods section.

Were not grow fungi or bacteria? 30 days in water? How do you affirm that it is from the plant extract and not fungi or bacteria compound? How was the yield of the extract? Have had concentrate on the extract?

Response

Many thanks for this really spot on comment. The detected compounds and their efficacy are presumably originating from the fruits/plants. This is confirmed by other studies that provided the information about the chemical analysis of the different parts of the fruits, such compounds were found in our macerate as well. It might be possible that the maceration process enhances the extraction or produces some by products. However, this might fall under the limit of detection of our chemical analysis. All the missing information has been updated in the manuscript, together with a sentence concerning the possible origin of the bioactive part of the macerates. We hope that this answer is now suitable for the reviewer.

Most references from Table 3 didn’t show anthelmintic properties from the compound. Extracts with this compound have been demonstrated with ATH activity. This is entirely different.

Response

Thanks for this comment. The references of the entire table have been updated.

Reviewer 2 Report

The manuscript was improved, I consider that can be published in this journal.

Author Response

The manuscript was improved, I consider that can be published in this journal.

Response

Many thanks to the referee for the great effort that helped to improve the manuscript.